# Balancing Utility and Scalability in Metric Differential Privacy

**Jacob Imola**[1]    **Shiva Prasad Kasiviswanathan**[2]    **Stephen White**[2]    **Abhinav Aggarwal**[2]    **Nathanael Teissier**[2]

[1] UC San Diego, La Jolla, CA,
[2] Amazon, USA

## Abstract

Metric differential privacy (mDP) is a modification of differential privacy that is more suitable when records can be represented in a general metric space, such as text data represented as word embeddings or geographical coordinates on a map. We consider the task of releasing elements of the metric space under metric differential privacy where utility is measured as the distance of the released element to the original element. Linear programming (LP) can be used to construct a mechanism that achieves the optimal utility for a particular mDP constraint. However, these LPs suffer from a polynomial explosion of variables and constraints that render them impractical for solving real-world problems. An important question is how to design rigorous mDP mechanisms that balance the utility-scalability tradeoff.

Our main contribution is a new method for reducing the LP size used to generate mDP mechanisms by constraining the search space such that certain input and output pairs have transition probabilities derived from the exponential mechanism. Our method produces mDP mechanisms whose LPs are smaller that all prior work in this area. We also provide a lower bound on the best possible mechanism utility. Our experiments on real-world metric spaces highlight the superior utility-scalability tradeoff of our mechanism.

## 1 INTRODUCTION

Privacy has emerged as a topic of strategic consequence across all computational fields. Differential privacy (DP), a mathematical formulation of privacy proposed by Dwork et al. [2006], provides provable protection guarantees against adversaries with arbitrary side information and computational power. See the book by Dwork and Roth [2013] for a primer on differential privacy and a survey of different techniques proposed in the literature.

More recently, researchers have noted that differential privacy does not take the underlying metric space of the data domain into account. Differential privacy provides the same level of protection to all perturbations of a single user's data which makes it inflexible when these perturbations are not all the same. For example, if the data consists of locations on earth, there is a large difference between discerning whether a user is in a 1 or a 100-mile radius. In many scenarios, the former type of privacy breach is more significant because a user's location is more accurately determined. This has led to the development of *metric* DP (mDP) which provides different protections depending on an underlying metric space, and has been adopted in applications involving releasing sensitive geolocation data [Andrés et al., 2013, Bordenabe et al., 2014] and textual data [Fernandes et al., 2019, Feyisetan et al., 2019, 2020, Xu et al., 2020, Feyisetan and Kasiviswanathan, 2021].

Mechanism utility for mDP is less well-understood than that of general DP, as the metric strongly influences the permitted behavior of the mechanism. While it is possible to design an optimal mechanism under mDP, it is also a computationally challenging task that requires solving a linear program (LP) with $O(n^2)$ variables and $O(n^3)$ constraints [Bordenabe et al., 2014], where $n$ is the size of the metric space (i.e., cardinality of the set). In fact, most mDP mechanisms [Feyisetan et al., 2020, 2019, Xu et al., 2020] do not provide any rigorous guarantees on the utility. Our key contributions are as follows:

**(a)** We present a general framework for designing mDP mechanisms which have a better tradeoff between mechanism utility and the size of the LP used to compute the mechanism (Section 3). This framework is based on adding new constraints to LP that certain transition probabilities are equal to those arising from a weighted version of the *exponential mechanism*.[1] We instantiate

---

[1] Exponential mechanism [McSherry and Talwar, 2007] is a

*Accepted for the 38th Conference on Uncertainty in Artificial Intelligence* (UAI 2022).

this framework, using new constraints based on $r$ nearest neighbors of each point, to produce an LP which has just $O(nr)$ variables and $O(n^2r)$ constraints. In practice, $r$ can be set as a small constant. Therefore, our new mechanism substantially increases the size of the metric space on which mDP mechanisms can be practically applied.

(b) We prove a lower bound on the word-level loss within the underlying metric space (Section 4). This provides the first non-trivial loss lower bound on any mDP mechanism, including the optimal one. This lower bound is valuable, especially in situations when the LP for the optimal mechanism is infeasible to solve, as it bounds the utility of any mDP mechanism. Our code is available online Imola [2022].

(c) We perform extensive experiments comparing the utility and privacy of our proposed mechanism and existing mechanisms in text and geolocation applications (Section 5). These experiments indicate that our proposed mechanism performs more closely to the optimal mechanism than others tested and can result in a utility improvement of about $25\%$ compared to the non-optimal mechanisms. In terms of scalability, our results indicate that our proposed mechanism can scale to metric spaces four times larger than the optimal mechanism.

**Related Work in Metric DP.** Metric DP originated in the context of location privacy where given a dataset of geolocation coordinates (longitude and latitude) on a plane, the notion of adjacency could be better captured using the Euclidean distance between the coordinates [Andrés et al., 2013, Chatzikokolakis et al., 2013]. Metric DP mechanisms have been investigated for various choices of metrics, including Euclidean, Manhattan, and Chebyshev metrics, among others [Chatzikokolakis et al., 2013, Andrés et al., 2013, Chatzikokolakis et al., 2015, Fernandes et al., 2019, Feyisetan et al., 2019, 2020]. Unlike our focus here, none of these results compare the loss of their proposed mechanisms to the optimal loss.

The most directly related work to ours is that of Bordenabe et al. [2014]. This paper proposes finding the optimal mDP mechanism using linear programming. They propose a method based on spanner graphs to reduce the size of the LP (outlined in full version). A $\delta$-spanner graph is a set of edges between points in a metric space such that the distance between two points in the graph approximates the metric up to a multiplicative factor $\delta$. Bordenabe et al. [2014] use a 3-spanner, for which a construction using just $O(n^{1.5})$ edges exists, to reduce the number of constraints in the LP from $O(n^3)$ to $O(n^{2.5})$.

**Related Work in Privately Releasing Text Embeddings.** Vector representations of words, sentences, and documents have all become basic building blocks in NLP pipelines and

algorithms. Hence, it is natural to consider privacy mechanisms that target these representations in the underlying metric space [Fernandes et al., 2019, Feyisetan et al., 2019, Xu et al., 2020, Feyisetan et al., 2020]. The most relevant result to our setting is the mechanism of Feyisetan et al. [2020] (referred to as *Madlib*). In Section 5, we compare our mechanism to Madlib as a baseline.[2]

# 2 TECHNICAL PRELIMINARIES

Throughout this paper, we consider data that comes from a finite metric space $(\mathcal{W}, d_{\mathcal{W}})$ where $\mathcal{W}$ is the set of values the data may take. For example, in the text release usecase, $\mathcal{W}$ consists of a vocabulary set, and in the geo-locations usecase, $\mathcal{W}$ consists of a set of locations. The metric $d_{\mathcal{W}} : \mathcal{W} \times \mathcal{W} \to \mathbb{R}$ captures dissimilarity between elements in the set. In NLP applications, it is very common to represent words via a high-dimensional text embedding $\phi : \mathcal{W} \to \mathcal{W}' \subseteq \mathbb{R}^d$.[3] Then we can define the distance between the words as the distance between the embedded words: i.e., for all $w_1, w_2 \in \mathcal{W}$, we define $d_{\mathcal{W}}(w_1, w_2) = d_{\mathcal{W}'}(\phi(w_1), \phi(w_2))$.

## 2.1 PRIVACY ON METRIC SPACES

Informally, a mechanism $\mathcal{M}$ satisfies metric DP[4] if its behavior is nearly the same on inputs that are close together in the metric space. This is formalized by the following notion of $\epsilon$-$d_{\mathcal{W}}$ privacy.

**Definition 1** (Metric DP (mDP)). *Given a finite set $\mathcal{W}$, a metric $d_{\mathcal{W}} : \mathcal{W} \times \mathcal{W} \to \mathbb{R}$, and a privacy parameter $\epsilon > 0$, a mechanism $\mathcal{M} : \mathcal{W} \to \mathcal{W}$ satisfies $\epsilon$-$d_{\mathcal{W}}$ privacy if for all $w_1, w_2, w \in \mathcal{W}$:*

$$\mathbb{P}r\left[\mathcal{M}(w_1) = w\right] \le \exp\left(\epsilon d_{\mathcal{W}}(w_1, w_2)\right) \mathbb{P}r\left[\mathcal{M}(w_2) = w\right].$$

The above definition is closely related to the definition of *local DP* [Kasiviswanathan et al., 2011] in that we apply $\mathcal{M}$ to each element of some database $D \in \mathcal{W}^m$ independently. The difference of mDP over local DP is that, because of the $d_{\mathcal{W}}$ term (which is absent in the local DP formulation), mDP mechanism guarantees indistinguishability for those $w_1, w_2 \in \mathcal{W}$ based on the distance $d_{\mathcal{W}}(\phi(w_1), \phi(w_2))$ between them. Similar to traditional differential privacy, mDP is preserved under post-processing

---

popular approach for deferentially private selection.

[2]While our mDP mechanism is applicable to any metric space, our experiments are over word embeddings and geolocations in the Euclidean space. Therefore, we do not directly compare with [Fernandes et al., 2019, Feyisetan et al., 2019, Xu et al., 2020] which work with embeddings in non-Euclidean spaces.

[3]Our results do not depend on the choice of the embedding.

[4]Metric DP is sometimes referred to as Lipschitz privacy [Koufogiannis et al., 2016], motivated by the fact that the privacy guarantee can be viewed as a Lipschitz condition on the mechanism, $|\ln(\mathbb{P}r\left[\mathcal{M}(w_1) = w\right]) - \ln(\mathbb{P}r\left[\mathcal{M}(w_2) = w\right])| \le \epsilon d_{\mathcal{W}}(w_1, w_2)$.

and composition of mechanisms [Koufogiannis et al., 2016]. In metric spaces, a natural definition of mechanism loss on an element $w \in \mathcal{W}$ is the expected distance between $w$ and $\mathcal{M}(w)$: $\mathcal{L}(\mathcal{M}, w) = \mathbb{E}_{\mathcal{M}}[d_{\mathcal{W}}(w, \mathcal{M}(w))]$. Here, the expectation is over the random bits in $\mathcal{M}$. We define the loss of $\mathcal{M}$ to be the worst-case loss of $\mathcal{M}$ on any particular element $w \in \mathcal{W}$:

$$\mathcal{L}(\mathcal{M}) = \max_{w \in \mathcal{W}} \mathcal{L}(\mathcal{M}, w) \qquad (1)$$

Notice that $\mathcal{L}(\mathcal{M})$ is non-negative due to $d_{\mathcal{W}}$ being a metric. Considering the loss as a worst-case instead of an average has the advantage that there cannot exist "adversarial" elements $w \in \mathcal{W}$ such that $\mathcal{L}(\mathcal{M}, w)$ is much higher than $\mathcal{L}(\mathcal{M})$. Similar loss functions have been studied in other DP settings such as in [Hardt and Talwar, 2010].

**Optimal Mechanism with LP.** It is easy to see that the constraints of mDP are linear. For a mechanism $\mathcal{M}$, we can consider its *stochastic matrix*[5] $M$ given by $M = \{M_{uv} : u, v \in \mathcal{W}\}$ with $M_{uv} = \Pr[\mathcal{M}(u) = v]$. Then, $\mathcal{M}$ satisfies mDP if and only if $M$ is stochastic and satisfies the following constraints

$$M_{uw} \leq M_{vw} \cdot \exp\left(d_{\mathcal{W}}(u, v)\epsilon\right) \qquad \forall u, v, w \in \mathcal{W} \quad (2)$$

Since the constraints are linear, mDP constrains $M$ to be in a polytope. We will overload notation and write $\mathcal{L}(M, w)$ and $\mathcal{L}(M)$ as the losses of the mechanism given by transition matrix $M$. These losses are given by:

$$\mathcal{L}(M) = \max_{u \in \mathcal{W}} \mathcal{L}(M, w)$$
$$= \max_{u \in \mathcal{W}} \sum_{v \in \mathcal{W}} d_{\mathcal{W}}(u, v)M_{uv}. \qquad (3)$$

Over the variables $M_{uv}$, $\mathcal{L}(\mathcal{M})$ is a maximum of linear functions. The optimal mechanism is given by the stochastic matrix $M$ that minimizes $\mathcal{L}(\mathcal{M})$ subject to the privacy constraints (2). Using standard techniques in linear programming, we can compute the best mechanism with the following LP over the variables $M, k$:

$$\mathcal{P}_{\text{OPTMECH}}(\epsilon) = \textbf{minimize } k \textbf{ subject to}$$
$$\mathcal{L}(M, w) \leq k, \ \forall w \in \mathcal{W}$$
$$M \text{ stochastic}$$
$$M \text{ satisfies (2)}$$

This LP problem has $O(n^2)$ variables and $O(n^3)$ constraints, where $n = |\mathcal{W}|$. Therefore, even with the state-of-the-art LP approaches, which all require $\Omega(N^2)$ time, where $N$ is the number of variables [Jiang et al., 2021], scalability is problematic (here $N = n^2$). This is the central motivation for our work.

---

[5]A stochastic matrix is a square matrix whose rows are probability vectors.

# 3 BALANCING UTILITY-SCALABILITY

Given the scalability issues in solving $\mathcal{P}_{\text{OPTMECH}}(\epsilon)$, a natural idea is to reduce the LP size. In this section, we present a new method to reduce the size of the LP in $\mathcal{P}_{\text{OPTMECH}}$ while still maintaining the mDP guarantee (Definition 1). Our method is based on adding *exponential mechanism* (ExPMECH) [McSherry and Talwar, 2007] equality constraints to the LP. Before we do this, we make use of an observation that the ExPMECH is provably not optimal in mDP which may be of independent interest. Thus, the constraints we add come from a "weighted" version of the ExPMECH. All missing proofs are collected in the full version.

## 3.1 IMPROVING THE ExPMECH IN MDP

Informally, the exponential mechanism [McSherry and Talwar, 2007] is a method for deferentially private selection from a discrete set of candidate outputs. Due to its flexibility the ExPMECH has become a popular tool for designing DP mechanisms. Furthermore, the ExPMECH is known to be optimal in DP for many choices of utility function [Hardt and Talwar, 2010, Aldà and Simon, 2017].

However, in the mDP setting, the exponential mechanism can be fooled by outlier elements. Informally, the dense areas of the metric space can act as a "black hole" where the ExPMECH will output elements in the dense area with high probability, even for the outlier elements. This drives up the loss for the outlier elements.

For a metric space $\mathcal{W} = \{w_1, \ldots, w_n\}$ and metric $d_{\mathcal{W}}$, the ExPMECH has the following transition probability

$$\Pr\left[\text{ExPMECH}(w_i) = w_j\right] = \frac{e^{-\epsilon d_{\mathcal{W}}(w_i, w_j)/2}}{\sum_{k=1}^{n} e^{-\epsilon d_{\mathcal{W}}(w_i, w_k)/2}}.$$

To illustrate on a concrete example, consider the metric space where $\mathcal{W} = \{w_1, \ldots, w_n\}$ and where $d_{\mathcal{W}}$ satisfies (1) $d_{\mathcal{W}}(w_1, w_i) = 1$ for $i \geq 2$, and (2) $d_{\mathcal{W}}(w_i, w_j) < \delta$ when $i, j \geq 2$, where $\delta$ is a small constant. For each $j \geq 2$, the ExPMECH satisfies $\Pr[\text{ExPMECH}(w_1) = w_j] = \frac{e^{-\epsilon}}{1 + ne^{-\epsilon}}$, and thus $\Pr[\text{ExPMECH}(w_1) \neq w_1] = \frac{ne^{-\epsilon}}{1 + ne^{-\epsilon}}$. As $n$ grows, the probability of this occurring approaches 1, and the loss does as well. The elements $w_i$ for $i \geq 2$ are acting as a "black hole".

This can be fixed by assigning weights to the outputs of the exponential mechanism. For positive weights $\mathbf{Y} = (Y_1, \ldots, Y_n) \in (\mathbb{R}^+)^n$, consider the weighted exponential mechanism given by

$$\Pr\left[\text{ExPMECH}_{\mathbf{Y}}(w_i) = w_j\right] = \frac{Y_j e^{-\epsilon d_{\mathcal{W}}(w_i, w_j)/2}}{\sum_{k=1}^{n} Y_k e^{-\epsilon d_{\mathcal{W}}(w_i, w_k)/2}}.$$

This mechanism can be shown to satisfy $\epsilon$-$d_{\mathcal{W}}$ privacy.

**Proposition 1.** *For any metric space and any* $\mathbf{Y} \in (\mathbb{R}^+)^n$, *the mechanism* EXPMECH$_\mathbf{Y}$ *satisfies* $\epsilon$-$d_\mathcal{W}$ *privacy.*

In our example, to avoid the problem encountered by the regular EXPMECH, we can weight $w_1$ higher than $w_2, \ldots, w_n$. For example, there exists a weighting $\mathbf{Y}$ such that the following loss is possible:

**Lemma 1.** *With* $\mathcal{W} = \{w_1, \ldots, w_n\}$ *and* $d_\mathcal{W}$ *defined as above, when* $\mathbf{Y} = (1, 1/(n-1), 1/(n-1), \ldots)$, *we have that*

$$\mathcal{L}(\text{EXPMECH}_\mathbf{Y}) \leq \frac{\frac{1}{n-1} + e^{-\epsilon/2}}{1 + e^{-\epsilon/2}} \mathcal{L}(\text{EXPMECH}).$$

*When* $\epsilon \geq 2\log n$, *we have* $\mathcal{L}(\text{EXPMECH}_\mathbf{Y}) \leq 2/(n-1)\mathcal{L}(\text{EXPMECH})$.

This establishes that the EXPMECH is provably not optimal on our example metric space. However, one problem with the more general EXPMECH$_\mathbf{Y}$ is that it is not clear how to set $\mathbf{Y}$ to optimize the loss other than the rule of thumb that dense elements should be weighted less. In the next section, we leave it to the LP solver to optimize these weights.

## 3.2 BALANCING LP LOSS AND SCALABILITY

To reduce the number of LP constraints required to find the optimal mechanism, our key idea is to add equality constraints in such a way that many of the original constraints in $\mathcal{P}_{\text{OPTMECH}}$ are trivially satisfied. This results in potentially a much smaller LP; however, optimality is no longer guaranteed. The balance between optimality and LP size is decided by the number of equality constraints. In fact, we will develop a general framework for balancing this tradeoff, which we call ConstOPTMech (Algorithm 1).

Specifically, to obtain the LP describing ConstOPTMech, we start with $\mathcal{P}_{\text{OPTMECH}}(\epsilon)$ and add non-negative variables $\{Y_w : w \in \mathcal{W}\}$. Then, for certain variables $M_{uv}$, we add additional "weighted exponential mechanism-like" constraints: $M_{uv} = Y_v e^{-\epsilon d_\mathcal{W}(u,v)}$. We leave the weights $Y_v$ to be optimized by the LP solver.

We allow deviations from the additional constraints in the form of a replacement function $I(v) : \mathcal{W} \rightarrow 2^\mathcal{W}$ that returns the elements $u \in \mathcal{W}$ for which the weighted exponential mechanism should not be used to set $M_{uv}$. To encode this, we add the following constraints:

$$M_{uv} = Y_v e^{-\epsilon d_\mathcal{W}(u,v)} \quad \forall u, v \in \mathcal{W}; u \notin I(v) \quad (4)$$

The replacement function $I(v)$ indicates where we do not want to use the exponential mechanism, and there are many candidates. We will later consider the following instantiation of the replacement function, based on $r$-nearest neighbors for a new parameter $r$:

$$I_{\text{NN},r}(v) = \{u \in \mathcal{W} | v \text{ is an } r\text{-nearest neighbor of } u.\}.$$
$$(5)$$

Intuitively, $I_{\text{NN},r}(v)$ returns the elements $u$ such that $v$ is one of the $r$ nearest neighbors of $u$. We employ this function $I_{\text{NN},r}$ because the exponential mechanism already assigns exponentially-low probabilities of returning the farthest elements from a given element, and we conjecture there is not much improvement to be made for such scenarios.

Adding the constraints (4) to $\mathcal{P}_{\text{OPTMECH}}(\epsilon)$ will satisfy the privacy constraints (2), but it may be impossible for $M$ to be stochastic. One can see this in the extreme example by setting $I(v) = \emptyset$; then (4) holds for all $u, v \in \mathcal{W}$, and non-negative assignment to $Y_v$ that makes $M$ stochastic need not exist. To fix this, we relax the constraint that $M$ be stochastic, and only insist that its rows sum to values more than 1. We add a penalization term involving a hyperparameter $\lambda > 0$ to the loss of each element to penalize the extent to which the rows sum to more than 1. As such, our loss function now takes the following form:

$$\tilde{\mathcal{L}}(M, w) = \sum_{u \in \mathcal{W}} M_{wu} d_\mathcal{W}(u, w) + \lambda \sum_{u \in \mathcal{W}} M_{wu} \quad (6)$$

With the modified loss function and relaxed stochasticity requirement, we obtain the LP giving ConstOPTMech.

$$\mathcal{P}_{\text{CONSTOPTMECH}}(\epsilon) = \textbf{minimize } k \textbf{ subject to}$$
$$\tilde{\mathcal{L}}(M, w) \leq k \;\; \forall w \in \mathcal{W}$$
$$M_{uv} \geq 0, \;\; \sum_{v \in \mathcal{W}} M_{uv} \geq 1 \;\; \forall u, v \in \mathcal{W}$$
$$M \text{ satisfies (2) and (4)}$$
$$Y_v \geq 0 \;\; \forall v \in \mathcal{W}$$

Notice that this LP is always feasible because one valid solution is $M_{uv} = Y_v e^{-\epsilon d_\mathcal{W}(u,v)}$ for all $u, v$—since there are no restrictions on the $Y_v$ variables, we can set them high enough so that $\sum_{v \in \mathcal{W}} M_{uv} \geq 1$.

The benefit of the equality constraints (4) is that we can drop a large number of the constraints in (2), as they are trivially satisfied. This allows us to find a solution to $\mathcal{P}_{\text{CONSTOPTMECH}}(\epsilon)$ much faster than $\mathcal{P}_{\text{OPTMECH}}(\epsilon)$.

**Theorem 1.** $\mathcal{P}_{\text{CONSTOPTMECH}}$ *is feasible, and it is possible to solve it using a linear program with* $n + 1 + \sum_{v \in \mathcal{W}} |I(v)|$ *variables and* $2n + \sum_{v \in \mathcal{W}} 2|I(v)|^2 + 3|I(v)|$ *constraints. The number of non-zero coefficients in the LP is at most* $2n^2 + \sum_{v \in \mathcal{W}} 2|I(v)|^2 + 5|I(v)|$.

Our choice to drop the stochasticity requirement of $M$ gave us feasibility of $\mathcal{P}_{\text{CONSTOPTMECH}}$, but the solution $M$ is no longer a mechanism because it is not stochastic. We obtain ConstOPTMech by simply normalizing the rows of $M$, forming a final stochastic matrix $H$. Any choice of the hyperparameter $\lambda$ gives rise to a valid $H$, and due to the normalization, the dependence of $H$ on $\lambda$ is non-linear. In

**Algorithm 1:** Mechanism ConstOPTMech

**Data:** Universe $\mathcal{W}$, metric $d_{\mathcal{W}}$, budget $\epsilon$, replacement function $I(v)$, hyperparameter $\lambda \in \mathbb{R}^+$.
**Result:** Transition matrix $H$.
$M, \text{loss} \leftarrow \texttt{Solve}(\mathcal{P}_{ConstOPTMech}(\frac{\epsilon}{2}))$ with $\lambda$;
**for** $u, v \in \mathcal{W}$ **do**
$\quad \left| \quad H_{uv} \leftarrow \frac{M_{uv}}{\sum_{w \in \mathcal{W}} M_{uw}}; \right.$
**return** $H$

practice, one can run ConstOPTMech for a set of values $\lambda$, and deploy the solution with the best loss.

Mechanism CONSTOPTMECH uses half of the privacy budget for solving $\mathcal{P}_{\text{CONSTOPTMECH}}(\frac{\epsilon}{2})$ because normalization may increase the privacy guarantee by a factor of 2. We are able to show the following privacy guarantee on CONSTOPTMECH. The proof is a generalization of Proposition 1.

**Theorem 2.** *For any set $\mathcal{W}$, metric $d_{\mathcal{W}}$, and budget $\epsilon$, replacement function $I(v) : \mathcal{W} \to 2^{\mathcal{W}}$, and hyperparameter $\lambda$, Mechanism* CONSTOPTMECH *satisfies $\epsilon$-$d_{\mathcal{W}}$ privacy.*

**Setting the Replacement Function.** The primary replacement function we consider is $I_{\text{NN},r}(v)$ (5), where $r$ is a hyperparameter dictating the number of nearest neighbors in (5). A smaller value of $r$ results in fewer LP constraints, while trading off optimality, as many of the transition probabilities will be fixed to the weighted exponential mechanism.

Specifically, in Theorem 1 when the replacement function $I(v) = I_{\text{NN},r}(v)$, then the number of variables in $\mathcal{P}_{\text{CONSTOPTMECH}}$ is at most $nr + n + 1$, the number of constraints is at most $n^2r + 3nr + 2n$, and the number of non-zero elements is at most $2n^2 + 5nr + 2n^2r$ (see full version).

As a special case, when we use $I(v) = I_{\text{NN},n-1}$, we add no equality constraints, and (4) becomes equivalent to (2). The LP size of $\mathcal{P}_{\text{CONSTOPTMECH}}$ is then $O(n^3)$, the same as that of $\mathcal{P}_{\text{OPTMECH}}$. When $r$ is a constant much less than $n$, then the number of variables is $O(nr)$ and the number of constraints is $O(n^2r)$, each term saving a factor of $n$. We note that $O(n^2r)$ is a worst-case bound that is not always tight. If one assumes that there is no $v \in \mathcal{W}$ such that at least $10r$ other elements of $\mathcal{W}$ count $v$ as an $r$-nearest neighbor, then $\sum_{v \in \mathcal{W}} |I(v)|^2 + |I(v)| \leq \sum_{v \in \mathcal{W}} 110r^2 \leq O(nr^2)$.

In Table 1, we compare the number of variables, constraints, and non-zero coefficients arising in $\mathcal{P}_{\text{CONSTOPTMECH}}$ as compared to those in the optimal mechanism ($\mathcal{P}_{\text{OPTMECH}}$) and the LP based of spanner graphs [Bordenabe et al., 2014] (referred to as $\mathcal{P}_{\text{SPANNERMECH}}$, see full version). We see that $\mathcal{P}_{\text{CONSTOPTMECH}}$ improves on all three of these quantities compared to the other two LPs when $r \ll n$. We note that these are worst-case upper bounds, and in practice the LP complexity measures may smaller. We perform detailed empirical analysis of the mechanisms in Section 5.

## 4 LOWER BOUNDS

In this section, we propose an easy to compute lower bound for mechanism loss. Our lower bound builds on the intuition that for an element $w \in \mathcal{W}$, if there are many elements that are far, but not too far, from $w$, then mDP forces the distribution $\mathcal{M}(w)$ to place significant mass on the elements which are farther away. This gives a lower bound on the loss of $\mathcal{M}$. To make this intuition formal, we define a packing of $\mathcal{W}$ to be a set of elements which are at least a certain distance from each other. In the following, let $B(x, r)$ denote the elements $y \in \mathcal{W}$ such that $d_{\mathcal{W}}(x, y) \leq r$.

**Definition 2.** *Let $\mathcal{W}$ be a set. A finite set $S \subseteq \mathcal{W}$ is called a $(c, r, Q)$-packing w.r.t. metric $d_{\mathcal{W}}$ if the following hold: $|S| = c$; for all $x, x' \in S$, $B(x, r) \cap B(x', r) = \emptyset$, i.e. the balls around the elements in $S$ of radius $r$ are disjoint; and for all $x, x' \in S$, $d_{\mathcal{W}}(x, x') \leq Q$, i.e. the maximum distance between any two elements in $S$ is at most $Q$.*

The lower bound we derive holds for any $(c, r, Q)$-packing of the metric space $\mathcal{W}$. The catch is that if a packing with a small $r$ or $c$ is used, the bound will not be strong. Our lower bound involves the quantity $N(w, S)$ that depends on a $w \in \mathcal{W}$ and a $(c, r, Q)$-packing $S$. $N(w, S)$ is given by:

$$N(w, S) = \sum_{s \in S} \exp(-d_{\mathcal{W}}(w, s)\epsilon),$$

We also have the lower bound $N(w, S) \geq 1 + (c - 1)\exp(-Q\epsilon)$ which follows because $S$ is a $(c, r, Q)$-packing. Notice that $N(w, S) \geq 1$ (because $w \in S$ and $d_{\mathcal{W}}(w, w) = 0$), and $N(w, S)$ grows linearly with the number of elements in $S$ and grows exponentially when the elements in $S$ are closer to $w$ or when $\epsilon$ decreases. This represents the increasing amount of mass that must be placed on these elements according to mDP. Our lower bound will grow stronger with increasing $N(w, S)$. The lower bound is as follows (proof in full version).

**Theorem 3.** *Consider an arbitrary set $\mathcal{W}$ and metric $d_{\mathcal{W}} : \mathcal{W} \times \mathcal{W} \to \mathbb{R}$. Then, for any $\epsilon > 0$, any mechanism $\mathcal{M}$ satisfying $\epsilon$-$d_{\mathcal{W}}$ privacy and any $(c, r, Q)$-packing $S$ of $\mathcal{W}$, it holds that*

$$\mathcal{L}(\mathcal{M}) \geq \max_{w \in \mathcal{W}} r \left(1 - \frac{1}{N(w, S)}\right). \qquad (7)$$

*It follows that*

$$\mathcal{L}(\mathcal{M}) \geq r \left(1 - \frac{1}{1 + (c - 1)\exp(-Q\epsilon)}\right). \qquad (8)$$

Both (7) and (8) have a simple interpretation. The $r$ term represents the minimum loss that must be incurred when a mechanism returns an element that is in a different ball than the starting ball in the packing. The $r$ term is multiplied by

| | # Variables | # Constraints | # Non-Zeroes |
|---|---|---|---|
| Optimal LP: $\mathcal{P}_{\text{OPTMECH}}$ | $O(n^2)$ | $O(n^3)$ | $O(n^3)$ |
| Spanner-based LP: $\mathcal{P}_{\text{SPANNERMECH}}$ [Bordenabe et al., 2014] | $O(n^2)$ | $O(n^{2.5})$ | $O(n^{2.5})$ |
| Our Method: $\mathcal{P}_{\text{CONSTOPTMECH}}$ with function $I_{\text{NN},r}$ | $O(nr)$ | $O(n^2 r)$ | $O(n^2 r)$ |

Table 1: Comparison of the number of variables and constraints for the various LP-based methods achieving mDP. Note that $\mathcal{P}_{\text{CONSTOPTMECH}}$ improves on existing methods when $r \ll n$.

$P = 1 - \frac{1}{N(w,S)}$, which can be interpreted as a probability since it is between 0 and 1 (since $N(w,S) \geq 1$). As we show in the theorem proof, $P$ is a lower bound on the probability that the mechanism returns an element in a different ball from $w$ and thus incurs the error $r$. $P$ increases with $N(w,S)$, which depends on the packing in the ways we identified above. $P$ is small only when $N(w,S)$ approaches 1, and using the bound $N(w,S) \geq 1 + (c-1)\exp(-Q\epsilon)$, we see the central term controlling its closeness to 1 is $(c-1)\exp(-Q\epsilon)$. Here the parameter $Q$ crucially comes into play because if the elements in the packing are too far apart, then mDP is a weak privacy guarantee, $N(w,S)$ will approach 1, and the lower bound will weaken.

An important special case of our theorem occurs when we take $S$ to be the two farthest elements $w_{max}^1, w_{max}^2 \in \mathcal{W}$. In this case, $S$ is a $(2, \frac{r^*}{2}, r^*)$-packing where $r^* = d_{\mathcal{W}}(w_{max}^1, w_{max}^2)$. Our lower bound then reads $\mathcal{L}(\mathcal{M}) \geq \frac{r^*}{2}\left(\frac{\exp(-r^*\epsilon)}{1+\exp(-r^*\epsilon)}\right)$.

## 5 EXPERIMENTAL RESULTS

We investigate through experiments how the loss of our proposed mechanism, ConstOPTMech, compares to other state-of-the-art mDP mechanisms.[6] We also include comparisons to our loss lower bound (derived in Section 4). Furthermore, we perform studies to compare ConstOPTMech to SpannerMech, as it is the most directly related work. To do this, we experimentally evaluate the complexity of solving the LPs used to compute both mechanisms. We conclude with a standalone ablation study on the hyperparameters $r, \lambda$ in ConstOPTMech to understand how they affect its loss and LP size in practice.

We focus on text embeddings and geolocation metric spaces because, as noted in Section 1, mDP mechanisms have primarily been used for privately releasing text and location data. We reiterate that our code is available online Imola [2022].

### 5.1 EXPERIMENTAL SETUP

Our experiments consist of running two types of experiments for sample metric spaces in both application domains.

The first experiment evaluates privacy vs. loss on a fixed metric space. The second evaluates scalability as the size of the metric space grows.

We measure utility (loss) of a mechanism based on (1). Since this loss is agnostic to any downstream modeling task performed on these private releases, we do not focus on any specific downstream task.

**Text Release Application:** For the text release application, the goal is to release words from a vocabulary privately, subject to mDP defined by a word metric. We let $\mathcal{W}$ consist of the set of English words, and we consider the word metric given by $d_{\mathcal{W}}(u,v) = d(\phi(u), \phi(v))$, where $\phi : \mathcal{W} \to \mathbb{R}^d$ is an embedding function and $d$ is the Euclidean distance. We used both the FastText [Bojanowski et al., 2017] and the GloVe embedding [Pennington et al., 2014] for our embedding $\phi$. Here, the loss function $\mathcal{L}(M)$ corresponds to how well mechanism $M$ preserves semantic meaning of words with respect to the word metric. While other loss functions may be considered depending on the downstream task, this is beyond the scope of this work.

The metric spaces for this application consist of sampled vocabularies $\mathcal{W}' \subseteq \mathcal{W}$. Instead of selecting $\mathcal{W}'$ at random, which would produce a set of completely unrelated words, we used a clustered approach to produce a more realistic vocabulary. Initially, $\mathcal{W}'$ consists of one random English word. To sample another word, we add a random word to $\mathcal{W}'$ with 50% chance. Otherwise we select a random word $w \in \mathcal{W}'$ and add one of $w$'s 50 closest English words according to $d_{\mathcal{W}}$. We repeat this process until $\mathcal{W}'$ has the desired size.

**Geolocation Release Application:** In the geolocation application, the goal is to release user location data, discretized into predetermined rectangular bins, respecting mDP and preserving the location as well as possible. The metric space consists of a set $\mathcal{W}$ of rectangular bins, and the metric $d_{\mathcal{W}}$ is the Euclidean distance between bin centers.

To generate the rectangular bins, we use the method of Bordenabe et al. [2014] who sample from the most popular regions in Beijing. Specifically, we start with the Geolife [Zheng et al., 2010] dataset, which consists of 17621 location traces in Beijing. We divide Beijing into rectangular bins of $0.005°$ (about 0.6 km) in width and height. For each trace, we consider its top 30 regions, and we form a histogram of top regions across all traces. To sample a

---

[6]In this section, we use ConstOPTMech to denote Algorithm 1 invoked with replacement function $I_{\text{NN},r}$.

metric space of size $n$, we take the $n$ most popular regions in the histogram.

**Performance Benchmarks:** We use the following benchmarks to measure mechanism privacy, loss, and scalability.

*Privacy:* In practice it is usually acceptable to use $(\epsilon, \delta)$-DP for some small $\delta$. We adopt $(\epsilon, \delta)$-mDP for our experiments, as we do not want to penalize an algorithm for having some small probability of two elements $u, v$ being distinguished. We say $M$ satisfies $(\epsilon, \delta)$-mDP if for any $u, v \in \mathcal{W}$ and $S \subseteq \mathcal{W}$, we have

$$\Pr[M(u) \in S] \leq e^{\epsilon d_{\mathcal{W}}(u,v)} \Pr[M(v) \in S] + \delta. \quad (9)$$

For a fixed $\delta$, we let $\epsilon_{\text{tight}}$ be the smallest $\epsilon$ such that $M$ satisfies $(\epsilon, \delta)$-MDP:

$$\epsilon_{\text{tight}}(M) = \inf_{\epsilon \geq 0} M \text{ satisfies } (\epsilon, \delta)\text{-mDP} \quad (10)$$

In our experiments, we set $\delta = 0.001$.

*Loss:* For practical considerations, we use a more robust measurement of loss in our experiments, where it may not be problematic if the mechanism performs poorly on a small fraction of elements. Instead of using the maximum loss over all elements $\mathcal{L}(M)$ (1), we use the $q$th-quantile over the set $\{\mathcal{L}(M, w) : w \in \mathcal{W}\}$:

$$\mathcal{L}_q(M) = \text{quantile}_q(\{\mathcal{L}(M, w) : w \in \mathcal{W}\}) \quad (11)$$

This loss estimate allows mechanisms to perform poorly on a small subset of the metric space, which in practice may be outlier or noisy data. In all experiments, we use $q = 95\%$ so that mechanisms are evaluated based on their losses on the best 95% of elements.

*LP Scalability:* To measure scalability of our mechanisms, we measured the time and number of nonzero coefficients (NNZ) used in the LPs. We use the number of nonzero coefficients over the number of variables or constraints since LP solvers tend to be optimized toward solving sparse LPs. We also consider computation time to be an important measure, as it captures complexity beyond the NNZ. For mechanisms that do not require linear programs, the computational requirements are trivial and we do not test them.

**Specific Details for Each Mechanism:** We tested five mechanisms: a) Our proposed ConstOPTMech, b) OPTMech (based on solving $\mathcal{P}_{\text{OPTMECH}}$), c) SpannerMech [Bordenabe et al., 2014], d) Madlib mechanism [Feyisetan et al., 2020], and e) EXPMECH [McSherry and Talwar, 2007]. Madlib, EXPMECH, and OPTMech have no further parameters other than $\epsilon$. For SpannerMech, we implement the algorithm as it is described in Bordenabe et al. [2014].

Mechanism ConstOPTMech takes $\lambda$ and $I(v)$ as parameters (see Algorithm 1). We optimize over $\lambda$ with possible values in $\{0.001, 0.1, 1.0\}$. For the replacement function $I(v)$, we

use $I_{\text{NN},r}(v)$ (5). We try both $r = 5$ and $r = 10$, and we will designate these values in our results.

**Evaluating Lower Bound:** For each metric space, we computed our lower bound according to Theorem 3. This theorem produces a lower bound for any $(c, r, Q)$ packing of $\mathcal{W}$ and any $\epsilon$. However, it is infeasible to try out every possible $(c, r, Q)$, packing. Instead, we generated candidate $(c, r, Q)$-packings using a $k$-center algorithm, using values of $k$ that varied from 1 to the size of the metric space. For each value of $\epsilon$ that we tested, we used the strongest lower bound given by one of our generated $(c, r, Q)$-packings.

**Experimental Outline:** The first experiments we conducted are *utility experiments*. We test which mechanisms are better at minimizing loss, subject to privacy constraints. To do this, we plot $\mathcal{L}_q$ versus $\epsilon_{\text{tight}}$ for a metric space consisting of 50 and 200 elements in both the text and geolocation release applications. We also plot the lower bound. For ConstOPT-Mech, we use $r = 10$. We do not run OPTMech for metric spaces of size 200, as the number of constraints would be $200^3$ which is too large. We do not include comparisons to Madlib for the Geolife dataset, as it is designed for text embeddings.

Next, we conduct *scalability experiments* on the mechanisms that involve solving LPs. We do this by fixing a privacy constraint $\epsilon_{\text{tight}}$ and, for each mechanism, testing the NNZ and time taken as the size of the metric space grows. We increased the number of samples in the metric space starting at 50 and increasing in increments of 50 until we reached 400 elements or the time spent solving the LP exceeded 1800 seconds. We fix $\epsilon_{\text{tight}} = 2.0$ for text release with the FastText embedding, at $1.0$ for text release with the GloVe embedding, and at $0.3$ for geolocation release. [7]

## 5.2 RESULTS

We discuss the experimental results for text release using the FastText embedding. The results for the GloVe embedding and the geolocation application are similar, and they appear in the full version.

**Utility Experiments:** Plots appear in Figure 1. In all tests, ConstOPTMech has lower loss than all other non-optimal mechanisms at all values of $\epsilon$. This includes high privacy regimes, where the loss is near the loss of returning a uni-

---

[7]In particular, the values of $\epsilon$ need not be the same across the metric spaces, as an mDP guarantee depends on both $\epsilon$ and the underlying metric space. Instead, we prioritize setting $\epsilon$ so that it is competitive with previous work while permitting low loss. Feyisetan et al. [2019] perform text release using the FastText and Glove embeddings with $\epsilon \in [5, 11]$, and Bordenabe et al. [2014] perform geolocation release using Geolife with $\epsilon \in [0.2, 2.0]$, and thus our values of $\epsilon$ are actually on the low end compared with previous work. Additionally, we will see in the utility experiments that our values of $\epsilon$ permit losses that are low with respect to the random baseline.

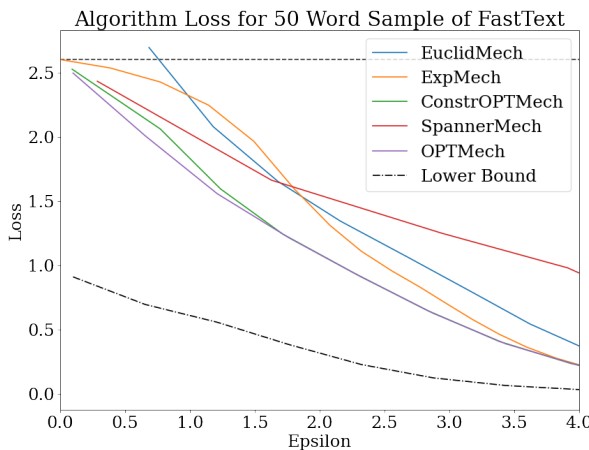 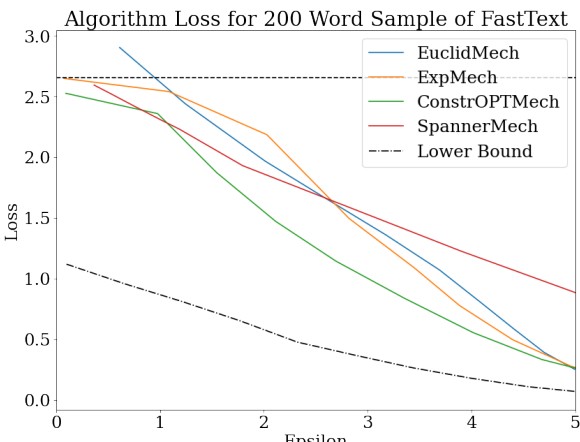

Figure 1: Loss of Madlib (EuclidMech), EXPMECH, ConstOPTMech, SpannerMech, and OPTMech versus $\epsilon_{\text{tight}}$ on 50 and 200-size metric spaces generated from FastText, along with the lower bound. The horizontal line indicates the loss of returning a uniform random element.

| Text (FastText) | 50 elements | 150 elements | 300 elements | 400 elements |
|---|---|---|---|---|
| ConstOPTMech (10) | 1.89 sec, 3.23e4 nnz | 24.91 sec, 1.47e5 nnz | 163.75 sec, 3.79e5 nnz | 234.63 sec, 5.71e5 nnz |
| ConstOPTMech (5) | 0.47 sec, 1.21e4 nnz | 12.85 sec, 6.60e4 nnz | 65.56 sec, 2.19e5 nnz | 141.90 sec, 3.70e5 nnz |
| SpannerMech | 1.25 sec, 2.12e4 nnz | 62.16 sec, 2.56e5 nnz | 1001.82 sec, 1.19e6 nnz | > 1800 sec, — nnz |
| OPTMech | 8.85 sec, 2.50e5 nnz | > 1800 sec, — nnz | | |

Table 2: Computation times and memory requirements for computing ConstOPTMech when $r = 10, 5$; SpannerMech; and OPTMech, for varying vocabulary sizes in the text release application using the FastText embedding. Each mechanism satisfies $\epsilon_{\text{tight}} = 2.0$.

formly random element, and low privacy regimes, where the loss approaches zero. The most pronounced improvement in loss occurs in middle ranges of $\epsilon$ (about $[1.0, 3.5]$) where ConstOPTMech offers an improvement of about 15-30% over all other non-optimal mechanisms. For example, when $\epsilon = 2$ on the 200-size vocabulary, ConstOPTMech offers a loss of 1.5, while the next-best mechanism, the EXPMECH, offers a loss of 1.8. This represents a 17% reduction, and a 44% reduction in the loss of returning a random word. This provides justification for setting $\epsilon = 2.0$ for the scalability experiments using FastText. The middle ranges of $\epsilon$ where ConstOPTMech is superior are the values with the most practical importance, since at these ranges, the losses are far from the random baseline yet are still nonzero—the mechanisms are offering both utility and privacy.

On the sampled metric spaces of size 50, ConstOPTMech attains only slightly worse loss than OPTMech, the optimal mechanism. As $\epsilon$ grows past 1.5, their losses become virtually the same. Because we are using $r = 10$, this means just $50 \times 10 = 500$ entries out of the 2500 entries in the transition matrix are not fixed. This suggests that the 10 nearest neighbors to an element play the largest role in minimizing the element's loss.

In all scenarios tested, there is a large gap between the lower bound and the losses of the mechanisms, even the optimal

mechanism. Hence, it is uncertain how close ConstOPT-Mech is to OPTMech on the metric spaces of size 200.

**Scalability Experiments:** We were able to run ConstOPT-Mech until 400 elements, whereas OPTMech timed out at 100 elements and SpannerMech timed out at 350 elements. Table 2 shows some of the time and NNZ data for the mechanisms. These results indicate that computing ConstOPTMech is faster than computing SpannerMech. This is particularly evident for the metric spaces with size 150 (resp. 300), where ConstOPTMech with $r = 10$ uses at most 40% (resp. 16%) as much time as SpannerMech, and ConstOPTMech with $r = 5$ uses at most 20% (resp. 6.5%) as much time. These faster times come despite running ConstOPTMech for three values of $\lambda$, which requires solving three LPs. In other words, the actual time to solve one LP used in ConstOPTMech is one third as high as the reported times.

In terms of NNZ, ConstOPTMech with $r = 10$ uses at most 57% (resp. 32%) as much time as SpannerMech, and ConstOPTMech with $r = 5$ uses at most 37% (resp 18%) as many non-zero coefficients on the metric spaces with size 150, 300. Note that one of the reasons why these savings are less than that observed for the time improvements is because ConstOPTMech uses LPs which are simpler than the LPs used by SpannerMech, which for example have

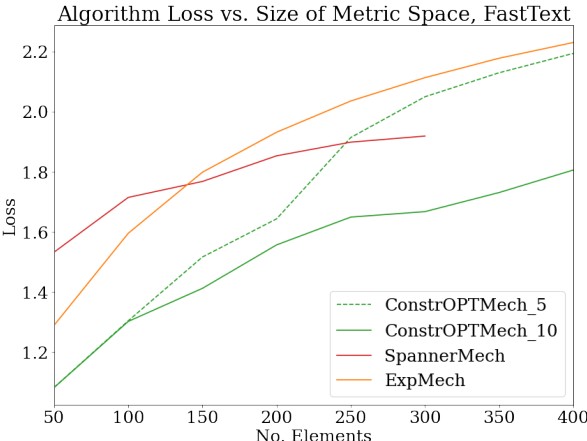

Figure 2: Loss of EXPMECH, ConstOPTMech (with $r = 5, 10$), and SpannerMech versus size of metric space for text release using the FastText embedding. Here, $\epsilon_{\text{tight}}$ is fixed at 2.0.

more variables ($O(n^2)$ variables versus $O(nr)$).

All the previous mechanisms have improved performance over OPTMech, which uses moderate time and NNZ on metric spaces with 50 elements and does not scale to 100 elements and beyond.

In Figure 2, we show our plots of loss versus the size of the metric space when $\epsilon_{\text{tight}}$ is fixed. The plot generally indicates that ConstOPTMech has lower loss than SpannerMech in addition to using a more scalable LP. At metric space sizes below 200, ConstOPTMech with $r = 10$ has approximately a 22% reduction in loss compared to SpannerMech, though this reduces to about 10% for metric space sizes greater than 200. ConstOPTMech with $r = 5$ outperforms SpannerMech by a margins on smaller metric spaces, though around 250 elements, ConstOPTMech with $r = 5$ begins performing much worse than when $r = 10$, and worse than even SpannerMech. However, using $r = 5$ with 250 elements means a very small fraction of nearest neighbors are not constrained. On large metric spaces, more nearest neighbors must be used to maintain low loss.

## 5.3 ABLATION STUDY

Finally, we conduct an ablation study on the hyperparameters $r, \lambda$ of ConstOPTMech, to see how they affect $\mathcal{L}_q$ and the LP scalability. To do this, we fixed the metric space to be a 200-word vocabulary using the FastText embedding, set $\epsilon_{\text{tight}} = 2.0$, and computed ConstOPTMech with a fixed $r$ and varying $\lambda$ while recording $\mathcal{L}_q$, the number of constraints in the LP, and the time spent computing ConstOPTMech. We then repeated the same with a fixed $\lambda$ and varying $r$.

The results of the ablation study appear in Table 3. When $r$ is fixed at 5 and $\lambda$ varies, the table indicates variation of about 6% in the losses achieved for the different values

| $\lambda$ | $\mathcal{L}_q$ | No. of constraints | Time (sec) |
|---|---|---|---|
| 0.100 | 1.64 | 107100 | 6.99 |
| 1.000 | 1.60 | 107100 | 6.70 |
| 10.000 | 1.56 | 107100 | 7.07 |
| 50.000 | 1.54 | 107100 | 7.87 |
| 250.000 | 1.54 | 107100 | 7.59 |
| $r$ | $\mathcal{L}_q$ | No. of constraints | Time (sec) |
| 1 | 2.04 | 81000 | 4.65 |
| 3 | 2.04 | 88472 | 5.98 |
| 5 | 1.54 | 107100 | 7.21 |
| 7 | 1.50 | 141404 | 7.90 |
| 9 | 1.49 | 186920 | 9.51 |

Table 3: Results of ablation study on hyperparameters $r, \lambda$. Here, the metric space is fixed to be a 200-word vocabulary with the FastText embedding, and $\epsilon_{\text{tight}} = 2.0$. In the first table, $r = 5$, and in the second table, $\lambda = 50$.

of $\lambda$, with the best choice of $\lambda$ being 50. As mentioned in Section 3, $\mathcal{L}_q$ has a non-linear dependence on $\lambda$, and this behavior is demonstrated in the table. There is little effect of $\lambda$ on the LP scalability.

When $\lambda$ is fixed at 50 and $r$ increases, $\mathcal{L}_q$ decreases by about 25% from its initial value of 1 to its final value of 9. However, this comes at the cost of an increased number of constraints in the LP, by about $2.3\times$, and a corresponding jump in the time required to solve the LP, by $2.1\times$. This demonstrates the better loss but increased LP complexity that comes with increasing $r$.

## 6 CONCLUSION

We tackle the problem of designing scalable metric differential privacy mechanisms that achieve near optimal utility. Our new mechanism combines the optimal LP-based mechanism and the exponential mechanism to achieve a better utility-scalability tradeoff than existing mechanisms. We also provide a simple to compute lower bound that improves our understanding of the optimal utility. Our experiments show that our mechanism is computationally tractable on larger metric spaces while also almost matching the utility of the optimal LP-based mechanism. While our mechanism operates on any metric space, an interesting question is whether the geometry of the metric space can be leveraged to improve either utility or scalability.

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
