# OpenReview forum: "Balancing Utility and Scalability in Metric Differential Privacy"
_auai.org/UAI/2022/Conference — UAI 2022 Poster_

### Official Review · Reviewer_QXDA · 2022-04-13

**Q2(1) Originality/Novelty:** 3
**Q2(2) Significance/Impact:** 3
**Q2(3) Correctness/Technical Quality:** 3
**Q2(6) Clarity Of Writing:** 2
**Q6 Overall Score:** 5
**Q8 Confidence In Your Score:** 3

**Q1 Summary And Contributions:**

This paper considers how to design the mDP mechanisms that balance the utility-scalability tradeoff. The main contribution of this paper is a new method to reduce the LP size to generate mDP mechanisms by constraining the search space.

**Q2 Assessment Of The Paper:**

More detailed information regarding each of these aspects is given below:

**Q2(4) Quality Of Experiments (Optional):**

3: Good: The experimental evaluation is adequate, and the results convincingly support the main claims.

**Q2(5) Reproducibility:**

2: Fair: Key resources (e.g., proofs, code, data) are unavailable but key details (e.g., proof sketches, experimental setup) are sufficiently well-described for an expert to confidently reproduce the main results.

**Q3 Main Strengths:**

This paper proposes a quite novel framework to make a tradeoff between utility and scalability for mDP mechanisms. It also proves a valuable lower bound. In addition, the experiments are quite comprehensive to prove the utility and salability.

**Q4 Main Weakness:**

There are still some points to be improved. For example, the authors should add more references to the experimental settings. In experiments, it’s better for showing more results about the parameter sensitivity. In addition, there’re some minor suggestions for improving the writing of this paper.

**Q5 Detailed Comments To The Authors:**

there still exist some points to be improved.
Here are some major comments:
1)	For the value of parameter settings in experiments, can you add more references about that? For example, “We fix … for metric spaces sampled from FastText, …at 1.0 for those sampled from …”.
2)	For the parameter r, besides setting the value as 5 and 10, can you show more results about how the parameter r affect the performance? For example, try more values about r, and show the performance.
3)	In experiments, it mentions that “we optimize over \lambda with possible values in {
0.001,0.1,1.0}”, can you show how the different values of \lambda affect the performance?
4) There’re still some grammar mistakes to be corrected. For example, it mentions “The plots confirms that …” which should be rewritten as “… confirm that…”. In the section conclusion, it seems better to replace the term “near optimal” as “near-optimal”.

**Q7 Justification For Your Score:**

See the above problems.

**Q9 Complying With Reviewing Instructions:**

1: Yes.

---

### Official Review · Reviewer_sVd6 · 2022-04-26

**Q2(1) Originality/Novelty:** 3
**Q2(2) Significance/Impact:** 3
**Q2(3) Correctness/Technical Quality:** 3
**Q2(6) Clarity Of Writing:** 3
**Q6 Overall Score:** 7
**Q8 Confidence In Your Score:** 2

**Q1 Summary And Contributions:**

To mitigate the scalability issues in addressing the original LP problem, the authors reduce the search space by proposing a weighted version of the exponential mechanism, which is technically sound. Moreover, the authors prove that the proposed mechanism satisfies \epsilon-d_{W} privacy. Built upon the adoption, the authors propose a framework to balance the tradeoff between optimality and scalability. To verify the effectiveness of the proposed method, extensive experiments are conducted.

**Q2 Assessment Of The Paper:**

More detailed information regarding each of these aspects is given below:

**Q2(4) Quality Of Experiments (Optional):**

2: Fair: The experimental evaluation is weak: important baselines are missing, or the results do not adequately support the main claims.

**Q2(5) Reproducibility:**

2: Fair: Key resources (e.g., proofs, code, data) are unavailable but key details (e.g., proof sketches, experimental setup) are sufficiently well-described for an expert to confidently reproduce the main results.

**Q3 Main Strengths:**

+ Developing scalable metric DP is interesting and promising, as the size of metric space can be large in many practical scenarios.

+ Introducing a weighted strategy for exponential mechanism-based mDP is novel and technical sound.

+ Theoretical analyses are given for the proposed method.


**Q4 Main Weakness:**

-  The experimental settings seem to be arbitrary. Detailed explanation for the utilized settings is required, i.e., following which work, as common settings can make the results more convincing.

-  Ablation study on the introduced hyper-parameter, e.g., \lambda, is missing.


**Q5 Detailed Comments To The Authors:**

- I suggest adding detailed explanations and the underlying intuitions of the proposed method. For example, why the introduced constraints in Eqn. (4) can ``encode'' the prior.

- I suggest adding empirical or theoretical analysis for the case where the proposed method gives a higher loss than baselines, i.e. when the number of elements is larger than 250 in Fig. 2.

**Q7 Justification For Your Score:**

I have read many papers on differential privacy.

**Q9 Complying With Reviewing Instructions:**

1: Yes.

---

### Official Review · Reviewer_mVNX · 2022-04-28

**Q2(1) Originality/Novelty:** 3
**Q2(2) Significance/Impact:** 2
**Q2(3) Correctness/Technical Quality:** 3
**Q2(6) Clarity Of Writing:** 2
**Q6 Overall Score:** 5
**Q8 Confidence In Your Score:** 3

**Q1 Summary And Contributions:**

The paper focuses on the problem of the semi-optimal MDP mechanism design considering the utility and scalability. By reducing the size of the LP maintained in the MDP mechanism problem, the authors propose a weighted LP joint renovated Expmech Shechem. In the proposed scheme, the utility and scalability tradeoff can be adjusted with corresponding constraints. The extensive experiments compared with existing scheme shows the proposed mechanism is more close to the optimal mechanism.

**Q2 Assessment Of The Paper:**

More detailed information regarding each of these aspects is given below:

**Q2(4) Quality Of Experiments (Optional):**

2: Fair: The experimental evaluation is weak: important baselines are missing, or the results do not adequately support the main claims.

**Q2(5) Reproducibility:**

2: Fair: Key resources (e.g., proofs, code, data) are unavailable but key details (e.g., proof sketches, experimental setup) are sufficiently well-described for an expert to confidently reproduce the main results.

**Q3 Main Strengths:**


+The idea of reducing the LP size to achieve the tradeoff between scalability and utility is interesting.
+The analysis and the derivation of the paper are well illustrated.
+The comparison of the existing works is extensive.

**Q4 Main Weakness:**

-The writing of the paper is not readable, especially as a theoretical analysis paper. The significance of the concepts is omitted, which makes the paper hard to understand. For instance, it implies the loss function of the mechanism indicates the quality of the designed mechanism, however, the above statement is not highlighted in Section2. Another example is that the paper indeed discusses the mDP design on finite sets, however, it is missing in the introduction sections.

-The position of the paper is not fully clear. In this paper, it compares the existing work on the experiments, but the differences are not discussed in the related work part. It should make a judgment on the existing works first and then compare the relative results.

-The significance of the lower bound is not well discussed. As the paper says, there is a large gap between the lower bound and all executed mechanisms in Section5. In this way, what's the meaning of such a loose lower bound? it is totally unclear.

**Q5 Detailed Comments To The Authors:**

The idea of the paper is interesting, but the readability of the paper is not strong. The authors should write the paper from the point of view of the readers and proofread the paper before the submission. All the figures are not readable since the transparency is too strong. The author can try to upsize the fonts in the figures.
For other comments, please see above.

**Q7 Justification For Your Score:**

Totally speaking, it is an interesting paper. The analysis of the paper is good, but the readability of the paper should be improved. The position and significance of the paper are not fully clear as it lacks sufficient discussion on the proposed and existing solutions.

I am wondering whether it should be published on UAI. Seems that the contribution to the AI field is limited as it has not checked the usability after the mDP noised mechanism.

**Q9 Complying With Reviewing Instructions:**

1: Yes.

---

### Decision · Program_Chairs · 2022-05-15

**Decision:**

Accept (Poster)

**Comment:**

Meta Review: This is a borderline paper on reducing the size of LP for solving the task of releasing elements of the metric space under metric differential privacy. The reviewers felt that the authors have done enough to merit acceptance but had several concerns:
1. Paper is not readable and therefore the general UAI audience may not appreciate its value.
2. Significance of the bounds is not discussed.
3. Discussion on the Impact of the various parameters used in the experiments

The authors have addressed all of these concerns in the rebuttal and the AC would like to suggest that they carefully address these points in the camera-ready version.